# Circulating Let-7 Family Members as Non-Invasive Biomarkers for Predicting Hepatocellular Carcinoma Risk after Antiviral Treatment among Chronic Hepatitis C Patients

**DOI:** 10.3390/cancers14082023

**Published:** 2022-04-16

**Authors:** Yi-Shan Tsai, Ching-I Huang, Pei-Chien Tsai, Ming-Lun Yeh, Chung-Feng Huang, Meng-Hsuan Hsieh, Ta-Wei Liu, Yi-Hung Lin, Po-Cheng Liang, Zu-Yau Lin, Shinn-Cherng Chen, Jee-Fu Huang, Wan-Long Chuang, Chia-Yen Dai, Ming-Lung Yu

**Affiliations:** 1Hepatobiliary Division, Department of Internal Medicine, Kaohsiung Medical University Hospital, Kaohsiung Medical University, Kaohsiung 80708, Taiwan; 1016ys@gmail.com (Y.-S.T.); tom65222@gmail.com (C.-I.H.); pctsai1225@gmail.com (P.-C.T.); yeh_ming_lun@yahoo.com.tw (M.-L.Y.); fengcheerup@gamil.com (C.-F.H.); hsmonyan@gmail.com (M.-H.H.); davyliu@gmail.com (T.-W.L.); 990076@kmuh.org.tw (Y.-H.L.); pocheng.liang@gmail.com (P.-C.L.); zuyali@kmu.edu.tw (Z.-Y.L.); chshch@kmu.edu.tw (S.-C.C.); jf71218@gmail.com (J.-F.H.); waloch@kmu.edu.tw (W.-L.C.); daichiayen@gmail.com (C.-Y.D.); 2Faculty of Internal Medicine, School of Medicine, College of Medicine, Kaohsiung Medical University, Kaohsiung 80708, Taiwan; 3Health Management Center, Kaohsiung Medical University Hospital, Kaohsiung Medical University, Kaohsiung 80708, Taiwan; 4Department of Occupational Medicine, Kaohsiung Medical University Hospital, Kaohsiung Medical University, Kaohsiung 80708, Taiwan; 5Center for Liquid Biopsy and Cohort Research, Kaohsiung Medical University, Kaohsiung 80708, Taiwan; 6Center for Infectious Disease and Cancer Research, Kaohsiung Medical University, Kaohsiung 80708, Taiwan; 7College of Professional Studies, National Pingtung University of Science and Technology, Pingtung 91201, Taiwan

**Keywords:** chronic hepatitis C, Let-7 family, hepatitis C virus, miRNA, biomarker, hepatocellular carcinoma, post-sustained virologic response

## Abstract

**Simple Summary:**

Cost-effective, Hepatocellular carcinoma (HCC) risk-based surveillance strategies should be established after achieving sustained virologic response (SVR). Circulating microRNAs are considered stable serum markers for early cancer diagnosis and prognosis and treatment response prediction. The aim of our longitudinal follow-up study was to assess whether Let-7 family members can predict HCC risk in CHC patients. We assessed the sera of 54 patients with CHC who developed HCC and 173 patients with CHC who did not develop HCC after antiviral therapy. Cox’s regression model revealed the independent role of let-7i as an effective surrogate biomarker for predicting HCC development in patients with CHC.

**Abstract:**

HCC, a leading cause of cancer-related mortality, is diagnosed at advanced stages. Although antiviral therapy has reduced the risk of HCC among chronic hepatitis C (CHC) patients, the risk of HCC remains, thus, highlighting the unmet need for continuous surveillance. Therefore, stable and cost-effective biomarkers, such as circulating microRNAs, must be identified. We aimed to clarify whether serum levels of the Let-7 family can predict HCC risk in patients with CHC using univariate and multivariate Cox’s proportional hazards model. We analyzed the sera of 54 patients with CHC who developed HCC after antiviral therapy and compared the data with those of 173 patients without HCC development. The Let-7 family (except for let-7c) exhibited significant negative correlations with the fibrosis score (*r* = −0.2736 to −0.34, *p* = 0.0002 to <0.0001). After Cox’s regression model was used to adjust for age, sex, HCV genotype, and FIB-4 ≥ 3.25, patients with CHC with let-7i median ≥ −1.696 (adjusted hazard ratio [aHR] = 0.31, 95% CI: 0.08–0.94, *p* = 0.0372) in the sustained virologic response (SVR) groups and ≥−1.696 (aHR = 0.09, 95% CI: 0.08–0.94, *p* = 0.0022) in the non-SVR group were less likely to develop HCC. Thus, circulating let-7i can be used for early CHC surveillance in patients with HCC risk after antiviral treatment.

## 1. Introduction

Chronic hepatitis C (CHC) virus infection is a major cause of advanced liver disease and hepatocellular carcinoma (HCC) worldwide. The annual HCC risk continues to be >2% for up to 10 years after the sustained virologic response (SVR) in CHC patients with cirrhosis and a Fibrosis-4 (FIB-4) score of ≥3.25 [1]. Hepatitis C (HCV)-infected patients have a 17-fold increased risk of developing HCC [2]. Cost-effective, HCC risk-based surveillance strategies should be established after SVR. HCV eradication with interferon and ribavirin therapy decreased the incidence of new-onset liver cancer in non-cirrhotic patients and the incidence of liver-related complications (decompensated liver disease or hepatocellular carcinoma) in cirrhotic patients with CHC [3]. Ultrasound (US)-based transient elastography, although widely available, low-cost, and user-friendly compared to computed tomography (CT) and magnetic resonance imaging (MRI), is not a sensitive predictor of early fibrosis [4]. Clinical markers, such as age, alanine aminotransferase, alpha-fetoprotein (AFP), and platelets, have been proposed to give a rough estimate of the risk of HCC. However, approximately 32–59% of patients with HCC have normal AFP levels; moreover, non-tumor-related AFP elevations may occur in patients with cirrhosis or chronic hepatitis, making AFP an inappropriate marker in surveillance [5].

MicroRNAs (miRNAs) are non-coding RNAs that post-transcriptionally modulate gene expression by affecting the stability and translation of complementary mRNAs. miRNAs repress essential HCV co-factors [6]. The members of the Let-7 family, miR-92, miR-122, miR-125b, mir-143, miR-192, miR-16, miR-126, and miR-199a/b have been found in normal livers and participate in the development of fibrogenesis and HCC [7,8]. The Let-7 family also includes 12 miRNA members, which are involved in the negative regulation of oncogenes and cell cycle regulators known as tumor suppressors [9,10]. The Let-7 family can be described by three distinct clusters of highly correlated expression in the circulating system [11]. Circulating Let-7 levels in plasma are correlated with hepatic fibrosis progression in CHC patients [12], and circulating microRNAs are considered stable serum markers for early cancer diagnosis, and to predict prognosis and response to therapy [13]. From systematic review and meta-analysis studies, it is clear that the circulation of specific aberrant miRNAs in the blood may increase the risk of HCC, and miRNAs could be utilized for the early diagnosis of HCC [14].

Interestingly, the role of the circulating Let-7 family members in HCC development in CHC patients after antiviral treatment has rarely been explored. Here, we conducted a longitudinal follow-up study with a well-characterized HCV cohort after antiviral therapy with and without SVR to determine the role of the circulating Let-7 family members in the prediction of HCC development.

## 2. Materials and Methods

### 2.1. Patient Cohort

Fifty-four patients with CHC who developed HCC after antiviral therapy (defined as the de novo HCC group, Appendix A) and 173 patients with CHC who did not develop HCC after antiviral therapy were enrolled in the study. The patients were positive for anti-HCV antibody for more than 6 months and were found to be positive for HCV RNA using polymerase chain reaction (PCR) assay. The patients achieved an SVR and showed undetectable HCV RNA after 24 weeks (M6) of peginterferon/ribavirin treatment. The exclusion criteria were a history of hepatitis B surface antigen, concomitant human immunodeficiency virus infection, other types of hepatitis, daily ethanol consumption ≥20 g (females and males), and no evidence of HCC before, during, or within 6 months of starting the antiviral therapy. FIB-4 was defined by the following equation: (age × AST [U/L])/(PLT [10^9^/L] × ALT [U/L]^1/2^). Patients with FIB-4 ≥ 3.25 were classified as having advanced fibrosis/cirrhosis [1]. The diagnosis of HCC was confirmed according to international guidelines such as spiral computed tomography (CT), magnetic resonance imaging (MRI), and biopsy [15,16]. The follow-up time was calculated from the date of anti-viral treatment at M6 to HCC diagnosis or the date of the last follow-up on 25 October 2021. HCV RNA and HCV genotype were assayed using a real-time PCR assay (RealTime HCV; Abbott Molecular, Des Plaines, IL, USA; detection limit, 12 IU/mL). All protocols were approved by the ethical committee of Kaohsiung Medical University Hospital based on the International Conference on Harmonization for Good Clinical Practice. The stored plasma was extracted from patients with CHC after they provided written informed consent from Kaohsiung Medical University between October 2009 and December 2016.

### 2.2. MicroRNA Extraction from Plasma

Circulating Let-7 family levels were measured at the initiation of antiviral treatment and 6 months after the end of treatment. Total RNA was extracted from 200 μL of plasma samples using 1 mL Trizol LS reagent and synthetic C. elegans Cel-39 as a spike-in control (Thermo Scientific, Wilmington, DE, USA), according to the manufacturer’s protocol as described previously [11].

### 2.3. Quantification of Circulating miRNAs

Two steps of Real-time PCR were performed on 20 ng of transcribed total RNA using the TaqMan MicroRNA Reverse Transcription kit and TaqMan^®^ Universal PCR Master Mix II, without uracil N-glycosylase (Thermo Fisher Scientific), according to the manufacturer’s instructions. TaqMan microRNA assays (assay ID): hsa-let-7a (000377), hsa-let-7b (002619), hsa-let-7c (000379), hsa-let-7d (002283), hsa-let-7e (002406), hsa-let-7f (000382), hsa-let-7g (002282), hsa-let-7i (002221), hsa-miR-98 (000577), and cel-miR-39-3p (000200) were used in the assay. All PCR reactions were performed on the 7900HT system (Applied Biosystems, Foster City, CA, USA) under the following conditions: 95 °C for 10 min, followed by 40 cycles of 95 °C for 15 s and 60 °C for 1 min. The relative expression level of each Let-7 member was determined using the comparative CT method, which was defined as 2^−ΔCt^, where ΔCt = Ct of the Let-7 member—Ct of cel-39 as described previously [11].

### 2.4. Statistical Analysis

Statistical analyses were performed using JMP 12.0 (SAS Institute, Cary, NC, USA). Graphs were generated using GraphPad Prism software (version 7.0; San Diego, CA, USA). Categorical variables and continuous variables were respectively calculated using chi-squared with Fisher’s exact test and Student’s *t*-test or ANOVA adjusted for multiple tests with Dunn-Bonferroni correction. The relationships between the Let-7 family members and different clinical parameters were analyzed using Pearson’s correlation coefficient analysis. Circulating Let-7 family levels were used to predict HCC development in CHC patients using Cox’s regression method and Kaplan-Meier survival analysis method. Statistical significance was set at *p* < 0.05.

## 3. Results

### 3.1. Characteristics of Patients with CHC

The mean time of HCC development after anti-viral treatment was 4.14 years (SD: 2.63), with patients having significantly higher age (57.87 ± 8.30 years, *p* = 0.0087), FIB-4 > 3.25 score proportion (*p* < 0.0001), AFP (0.0063), AST (*p* < 0.0001), ALT (*p* = 0.0008), and GGT levels (*p* = 0.0004), but lower HCV viral loads (*p* = 0.0494), and platelet count (*p* < 0.0001) than the group without HCC development (Table 1).

### 3.2. Comparison of Expression among Circulating Let-7 Family between Patients with and Those without HCC Development

In Table 2, we investigated the circulating Let-7 family twice, before (baseline) and after (M6) antiviral treatment regimen. There was no significant correlation between circulating Let-7 levels in the pre-antiviral treatment regimen. After antiviral treatment (M6), the circulating let-7a (*p* = 0.0011), let-7b (*p* = 0.0015), let-7g (*p* = 0.0025), and let-7i (*p* <0.0001) levels were significantly lower than the group without HCC development. We also measured the dynamic change (Δ) in circulating Let-7 levels as post antiviral treatment value (post; M6) minus baseline (pre; baseline) value. After antiviral treatment, there was a significantly lower dynamic change (Δ) in let-7g (*p* = 0.0025) and let-7i (*p* = 0.0013) in the de novo HCC group than in the group without HCC development.

### 3.3. Correlation Analysis between Let-7 Family Members and Laboratory Parameters

Pairwise correlations of Let-7 family members and laboratory parameters (Table 3), showed a meaningful negative correlation between let-7a (*r* = −0.1594, *p* = 0.0272), let-7e (*r* = −0.163, *p* = 0.0239), let-7f (*r* = −0.1456, *p* = 0.0439), and miR-98 (*r* = −0.1971, *p* = 0.0062) with AST were observed. Furthermore, let-7b (*r* = −0.2318, *p* = 0.0116), let-7d (*r* = −0.1943, *p* = 0.0116), let-7i (*r* = −0.2439, *p* = 0.0078), and miR-98 (*r* = −0.2242, *p* = 0.0147) were negatively correlated with GGT. We also observed meaningful positive correlations between platelet count and Let-7a (*r* = 0.2989, *p* < 0.0001), let-7b (*r* = 0.3171, *p* < 0.0001), let-7d (*r* = 0.4028, *p* < 0.0001), let-7e (*r* = 0.3096, *p* < 0.0001), let-7f (*r* = 0.2833, *p* < 0.0001), let-7g (*r* = 0.2595, *p* < 0.0001), let-7i (*r* = 0.3847, *p* < 0.0001), and miR-98 (*r* = 0.3015, *p* < 0.0001), but not let-7c. A negative correlation between FIB-4 score and Let-7 family members (except for let-7c) was also observed. Circulating Let-7 family members were significantly negatively correlated with a FIB-4 score of <3.25 (Appendix A).

### 3.4. Let-7 Family Performance for Prediction of HCC Development in CHC Patients after Anti-Viral Treatment

We performed univariate Cox regression analysis and found that let7b (median: ≥−1.44 vs. <−1.44, crude HR = 0.48, 95% CI: 0.21–0.67, *p* = 0.0007), let7d (median: ≥−1.90 vs. <−1.90 crude HR = 0.35, 95% CI: 0.18–0.61, *p* = 0.0002), let7i (median: ≥−1.696 vs. <−1.696 crude HR = 0.22, 95% CI: 0.18–0.61, *p* < 0.0001), dynamic change (Δ) in let-7d levels (median ≥0.17 vs. <0.17 crude HR = 0.42, 95% CI: 0.23–0.73, *p* = 0.0022), and Δlet-7e levels (median: ≥0.15 vs. <0.15 crude HR = 0.37, 95% CI: 0.20–0.65, *p* = 0.0005) were significant predictors of HCC. We stratified the patients by SVR status and adjusted by age, sex, HCV genotype, FIB-4 ≥3.25, only let7b (median: ≥−1.44 vs. <−1.44, adjusted HR = 0.30, 95% CI: 0.08–0.89, *p* = 0.0296) and let7i (median: ≥−1.696 vs. <−1.696, adjusted HR = 0.31, 95% CI: 0.08–0.94, *p* = 0.0372) (Table 4). In the non-SVR group, let7i (median: ≥−1.696 vs. <−1.969, adjusted HR = 0.09, 95% CI: 0.01–0.45, *p* = 0.0022) showed an 11-fold decreased cumulative probability of HCC development (Figure 1). Together, the findings suggest that let-7i is an independent factor for HCC development in patients with CHC.

## 4. Discussion

In this study, we investigated the circulating levels of let-7a/b/g/i and found them to be significantly lower in the HCC development group after antiviral treatment. Let-7 is one of the multiple miRNAs that regulate the infection process, becoming downregulated in HCV infection [6,17]. After antiviral treatment, we found that the circulating Let-7 family was higher in patients with CHC with SVR than in non-SVR (Appendix A), but it was still suppressed in CHC patients with HCC development than those without HCC development (Appendix A). Let-7 family members in CHC patients belonging to Cluster 1 include let-7a/d/e/g, members in Cluster 2 comprise of let-7b and let-7i, and those in Cluster 3 are let-7c/f/miR-98 [11]. The Kaplan-Meier survival analysis of the let-7 family cluster 1(let-7a, let-7d, let-7e, and let-7g) and cluster 3 (let-7c, let-7f, and miR-98) with HCC development in the SVR and non-SVR groups are shown in Appendix A, respectively. Thus, cluster 2 may be stable surveillance of blood biomarkers in liver cirrhosis and HCC development. In this study, we stratified CHC patients with an SVR status and adjusted for age, sex, log HCV RNA, HCV type, and LC (FIB-4 > 3.25) as covariables (Table 4 and Appendix A); let-7i was a useful marker for discriminating CHC patients with HCC development after antiviral treatment. Let-7 (a/d/e/g) expression positively correlated with PLT as HCV infection [11]. After anti-viral treatment, we found that the Let-7 family member significantly, negatively correlated with AST and hepatic fibrosis score (FIB-4). The serum let-7a level is superior to M2BPGi, FIB-4, and APRI and is comparable to liver stiffness in discriminating liver cirrhosis [18]. Moreover, the measurement of let-7a, let-7c, and let-7d levels in extracellular vesicles has no advantage over the direct measurement in plasma [12].

HCV-induced oncogenic Lin28B expression was observed by Ali et al. [19]. The human Lin28 family is composed of two homologs, namely Lin28 (also known as Lin28A and Lin28B). Lin28 is specifically expressed in undifferentiated embryonic stem cells. Lin28A/B can recognize two distinct regions of the RNA and inhibits Let-7 in vivo [20]. Lin28A/B functions as a reprogramming factor together with Oct4, Nanog, and Sox2, to regulate the pluripotency of stem cells [21]. Let-7 represses Lin28 expression through its 3′UTR, promotes differentiation, and inhibits self-renewal during the transition of *Caenorhabditis elegans* larvae to adulthood [22,23]. Lin28 promotes transformation and is associated with advanced human malignancies [24]. Lin28B overexpression has been noted in most HCC cell lines and clinical samples [25].

In contrast, a previous study showed that let-7b has a significant anti-HCV effect [26] and interferon-alpha (IFNα) rapidly modulates the expression of let-7s with anti-hepatitis C virus activity by targeting IGF2BP1 [27]. Furthermore, let-7g may act as a tumor suppressor gene that inhibits HCC cell proliferation by downregulating c-Myc and upregulating the tumor suppressor gene p16 (INK4A) [28]. Loss of let-7 enhances oncogene *RAS* and *HMGA2* expression, which also promotes ”stemness” by repressing self-renewal and promoting differentiation in tumor-initiating cells [29].

Chronic inflammation is a key promoter of cancer. Almost 30% of cancers have been attributed to chronic infections, 30% to environmental factors, and 35% to dietary factors [30]. It is noteworthy that Let-7 can directly repress TLR4 through 3′UTR post-transcriptional regulation, reducing NF-κB activity (Teng et al., 2013). Overexpression of TLR4 in CHC patients is correlated with the inflammatory score and degree of fibrosis [31]. TLR4 plays an important role in the pathogenesis of HCV-related chronic liver disease [32]. Let-7 also directly inhibits IL-6 expression through Ras and NF-κB pathways to prevent cell transformation [33]. Thus, Let-7 regulates chronic inflammation to prevent tumorigenesis.

This study had several limitations. It is known that the combination of AFP, AFP-L3, and DCP had greater predictive power than any individual marker. AFP levels were not available after anti-viral treatment. DCP was not routinely used in Taiwan before 2020. AFP-L3 is not available clinically in Taiwan. Therefore, we cannot compare the advantages and disadvantages of surrogate biomarkers. However, SVR rates of interferon-alpha treatment with immune modulation effects were lower than those of direct antiviral agents, which have 95–99% SVR rates [34], and have been permitted by the Taiwan National Health Insurance Administration Ministry of Health and Welfare since 2017. Even though post-DAA-SVR HCC development patients are difficult to obtain, a high fibrosis stage is also significantly associated with an elevated risk of HCC onset [35]. The current study demonstrated that Let-7 was significantly negatively correlated with the FIB-4 score in the FIB-4 score <3.25 group. Let-7 directly represses TGF-β1 receptor 1 (TGFBR1) expression in the kidney [36], suggesting that Let-7 regulates fibrogenesis in the liver.

## 5. Conclusions

The main findings in the study were that the let-7i levels are early predictors of HCC development at post-anti-viral treatment and that circulating let7i levels may be of use in the early surveillance of CHC with HCC risk.

## Figures and Tables

**Figure 1 cancers-14-02023-f001:**
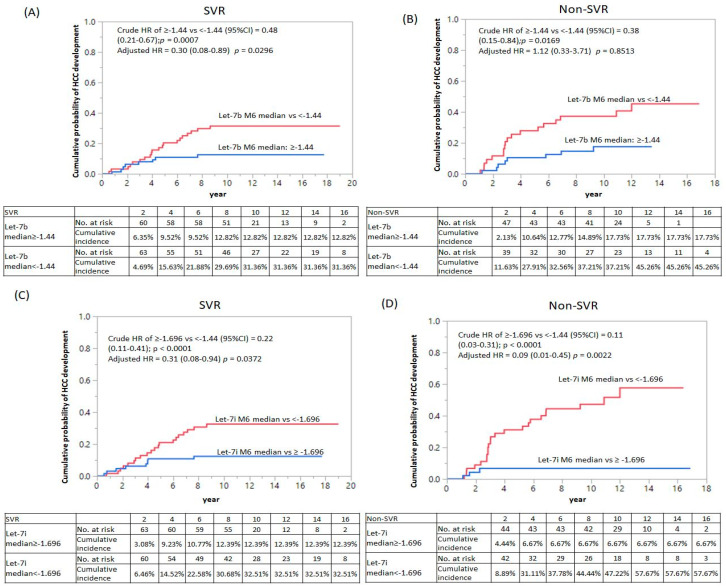
Kaplan-Meier survival analysis of let-7b and let-7i with HCC development in SVR and non-SVR group. Comparison of the cumulative probability of HCC development divided cut-off value from the median value of the distribution of Let-7 expression based on Log_10_ 2^−delta Ct method in (**A**) let-7b in SVR group, (**B**) let-7b in the non-SVR group, (**C**) let-7i in SVR group, and (**D**) let-7i in the non-SVR group. HR: Hazard ratio; CI: Confidence interval; HCC: Hepatocellular carcinoma.

**Table 1 cancers-14-02023-t001:** Participant CHC patients’ characteristics.

Parameters	Total (*n* = 227)	Without HCC Development (*n* = 173)	De Novo HCC (*n* = 54)	*p*
Age (yr, mean ± SD)	55.15 ± 8.75	54.31 ± 8.73	57.87 ± 8.30	0.0087
Gender (F/M)	116/111	93/80	23/31	0.1515
BMI	25.33 ± 3.60	25.40 ± 3	25.30 ± 3.05	0.9484
Total Viral loads log_10_ (IU/mL) #	5.62 ± 0.85	5.68 ± 0.85	5.42 ± 0.81	0.0494
HCV genotype (type1/non-type1/unclassified)	(123/99/5)	(95/76/2)	(28/23/3)	0.2510
FIB-4 > 3.25 (yes/no)	(114/122)	(67/106)	(37/16)	<0.0001
SVR (Yes/No)	(130/97)	(102/71)	(28/26)	0.4924
AFP (ng/mL)	22.88 ± 55.43	16.91 ± 37.02	41.62 ± 89.95	0.0063
Cr	0.83 ± 0.25	0.81 ± 0.23	0.88 ± 0.31	0.0609
AST ± SD (IU/L)	90.36 ± 53.72	81.37 ± 43.58	119.72 ± 71.12	<0.0001
ALT ± SD (IU/L)	123.04 ± 69.95	114.41 ± 59.71	150.70 ± 0.94	0.0008
GGT (IU/L)	68.16 ± 66.36	58.82 ± 57.99	96.17 ± 81.12	0.0003
WBC	5437.45 ± 1639.53	5503.24 ± 1722.15	5226.67 ± 1333.40	0.2946
Platelet count (×10^3^/µL, mean ± SD)	153.20 ± 61.61	162.88 ± 63.07	122.20 ± 44.68	<0.0001
Hemoglobin (g/dL)	14.03 ± 1.47	14.09 ± 1.50	13.85 ± 1.36	0.3457
Follow-up period (year; mean ± SD)	9.30 ± 4.18	10.98 ± 3.06	4.14 ± 2.63	<0.0001
Ct valube_pre_Cel-39	27.16 ± 1.03	27.12 ± 1.09	27.30 ± 0.80	0.3629
Ct valube_post_Cel-39	27.25 ± 1.13	27.20 ± 1.11	27.39 ± 1.20	0.2533

BMI, body mass index; HCC, hepatocellular carcinoma; LC, liver cirrhosis; Cr, creatinine; AST, aspartate aminotransferase; ALT, alanine aminotransferase; GGT, gamma-glutamyl transferase; WBC, white blood cell count; PLT, platelet count. All values are expressed as the mean ± standard deviation (SD). The *p*-value was calculated for the continuous variables using the Student’s *t*-test or Mann–Whitney test, and the χ^2^ test was used for the categorical variables, with α = 0.05. # The HCV virus loads were determined by log transformation.

**Table 2 cancers-14-02023-t002:** Circulating Let-7 family expression at baseline and after anti-viral treatment (M6).

Log _10_2^−Delta Ct	Procedure	Without HCC Development (*n* = 173)	De Novo HCC (*n* = 54)	*p*
let7a	Pre	−1.89 ± 0.59	−1.98 ± 0.50	0.2951
	Post	−1.65 ± 0.67	−1.97 ± 0.45	0.0011 *
	Δ post−pre	0.24 ± 0.73	0.01 ± 0.59	0.0405
let7b	Pre	−1.55 ± 0.59	−1.55 ± 0.61	0.9684
	Post	−1.31 ± 0.73	−1.66 ± 0.54	0.0015 *
	Δ post−pre	0.24 ± 0.82	−0.11 ± 0.77	0.0062
let7c	Pre	−2.17 ± 0.44	−2.20 ± 0.38	0.6392
	Post	−1.94 ± 0.48	−1.77 ± 0.95	0.0787
	Δ post−pre	0.23 ± 0.61	0.43 ± 1.04	0.0726
let7d	Pre	−1.90 ± 0.52	−1.99 ± 0.44	0.2448
	Post	−1.66 ± 0.66	−2.04 ± 0.39	0.0078
	Δ post−pre	0.23 ± 0.72	−0.05 ± 0.48	0.0092
let7e	Pre	−1.68 ± 0.64	−1.68 ± 0.56	0.9623
	Post	−1.48 ± 0.71	−1.72 ± 0.85	0.0557
	Δ post−pre	0.18 ± 0.80	−0.04 ± 1.00	0.0979
let7f	Pre	−2.17 ± 0.42	−2.25 ± 0.30	0.2065
	Post	−2.04 ± 0.53	−2.20 ± 0.36	0.0353
	Δ post−pre	0.13 ± 0.57	0.04 ± 0.47	0.3212
let7g	Pre	−1.90 ± 0.57	−1.83 ± 0.55	0.4272
	Post	−1.68 ± 0.67	−1.95 ± 0.46	0.0053 *
	Δ post−pre	0.22 ± 0.73	−0.12 ± 0.69	0.0025 *
let7i	Pre	−1.80 ± 0.57	−1.83 ± 0.59	0.7959
	Post	−1.53 ± 0.68	−1.96 ± 0.52	<0.0001 *
	Δ post−pre	0.28 ± 0.80	−0.13 ± 0.81	0.0013 *
miR−98	Pre	−2.29 ± 0.34	−2.29 ± 0.26	0.8889
	Post	−2.17 ± 0.44	−2.29 ± 0.41	0.0803
	Δ post−pre	0.12 ± 0.52	−0.003 ± 0.48	0.1180

Data are presented as the mean ± SD. The change (Δ) denotes post antiviral treatment (post; M6) minus baseline (pre; baseline) value of Let-7 level based on Log _10_2^-delta Ct (Ct _target_—Ct_cel39)_ method. * = *p* < 0.05 following statistical analysis using an ANOVA with Bonferroni correction (α = 0.0056).

**Table 3 cancers-14-02023-t003:** Pairwise correlation coefficients were calculated for the Let-7 family and laboratory parameters at post-anti-viral treatment (M6).

Log _10_2^−Delta Ct	AST	*p*	ALT	*p*	GGT	*p*	PLT	*p*	FIB−4 Score	*p*
let7a	−0.22 *	0.0027	−0.1476*	0.0317	−0.1587	0.0946	0.317 *	<0.0001	−0.3088 *	<0.0001
let7b	−0.116	0.1176	−0.0334	0.6287	−0.2202 *	0.0197	0.3357 *	<0.0001	−0.3108 *	<0.0001
let7c	−0.123	0.0975	−0.1062	0.1233	0.0623	0.5143	0.0602	0.382	−0.0414	0.5797
let7d	−0.182 *	0.0134	−0.1123	0.1031	−0.1846	0.0513	0.4361 *	<0.0001	−0.3644 *	<0.0001
let7e	−0.206 *	0.005	−0.1126	0.1019	−0.0648	0.4974	0.3336 *	<0.0001	−0.3391 *	<0.0001
let7f	−0.183 *	0.0129	−0.1017	0.1399	−0.1486	0.1178	0.3247 *	<0.0001	−0.3144 *	<0.0001
let7g	−0.156 *	0.0348	−0.067	0.3313	−0.1136	0.2332	0.2838 *	<0.0001	−0.3035 *	<0.0001
let7i	−0.162 *	0.028	−0.1127	0.1017	−0.2294 *	0.015	0.3915 *	<0.0001	−0.3714 *	<0.0001
miR98	−0.209 *	0.0044	−0.1553*	0.0237	−0.1972 *	0.0372	0.3238 *	<0.0001	−0.2736 *	0.0002

* Statistically significant at *p* ≤ 0.05; AST, aspartate aminotransferase; ALT, alanine aminotransferase; GGT, gamma-glutamyl transferase.

**Table 4 cancers-14-02023-t004:** Let-7 family associated with the development of HCC in the CHC patients after antiviral treatment (M6): Univariate and Multivariate Cox proportional hazard regression.

Log _10_2^−Delta Ct	Procedure	HR(95%CI)	*p*	Adjusted HR (95%CI)	*p*
let7a	M6 median: ≥−1.88 vs. <−1.88	0.47 (0.26–0.81)	0.0067	0.68 (0.25–1.80)	0.4490
	Δ post(M6)-pre(baseline) median: ≥0.18 vs. <0.18	0.50 (0.28–0.86)	0.0128	0.42 (0.16–1.06)	0.0669
let7b	M6 median: ≥−1.44 vs. <−1.44	0.48 (0.21–0.67)	0.0007 *	0.30 (0.08–0.89)	0.0296 *
	Δ post(M6)-pre(baseline) Median: ≥0.14 vs. <0.14	0.70 (0.40–1.20)	0.1959	0.93 (0.37–2.31)	0.8752
let7c	M6 median: ≥−2.02 vs. <−2.02	1.02 (0.59–1.74)	0.9521	1.62 (0.57–5.32)	0.3763
	Δ post(M6)-pre(baseline) median: ≥0.198 vs. <0.198	1.16 (0.68–2.00)	0.5809	1.72 (0.64–5.44)	0.2878
let7d	M6 median: ≥−1.90 vs. <−1.90	0.35 (0.18–0.61)	0.0002 *	0.90 (0.29–2.78)	0.8591
	Δ post(M6)-pre(baseline) median: ≥0.170 vs. <0.170	0.42 (0.23–0.73)	0.0022	0.62 (0.22–1.66)	0.3405
let7e	M6 median: ≥−1.67 vs. <−1.67	0.47 (0.26–0.82)	0.0072	1.25 (0.43–3.57)	0.6721
	Δ post(M6)-pre(baseline) median: ≥0.150 vs. <0.150	0.37 (0.20–0.65)	0.0005 *	0.55 (0.20–1.43)	0.2259
let7f	M6 median: ≥−2.16 vs. <−2.16	0.69 (0.40–1.18)	0.1778	1.84 (0.67–5.35)	0.2395
	Δ post(M6)-pre(baseline) median: ≥0.0918 vs. <0.0918	0.83 (0.48–1.41)	0.4954	1.42 (0.55–3.85)	0.4658
let7g	M6 median: ≥−1.91 vs. <−1.91	0.58 (0.33–1.00)	0.0537	0.95 (0.35–2.49)	0.9208
	Δ post(M6)-pre(baseline) Median: ≥0.12 vs. <0.12	0.60 (0.34–1.04)	0.0690	1.34 (0.52–3.57)	0.5473
let7i	M6 median: ≥−1.696 vs. <−1.696	0.22 (0.11–0.41)	<0.0001 *	0.31(0.08–0.94)	0.0372 *
	Δ post(M6)-pre(baseline) median: ≥0.177 vs. <0.177	0.53 (0.30–0.92)	0.0237	1.08 (0.41–2.83)	0.8793
miR-98	M6 median: ≥−2.235 vs. <−2.235	0.85(0.49–1.45)	0.5538	1.81 (0.69–5.08)	0.2293
	Δ post(M6)-pre(baseline) median: ≥0.069 vs. <0.069	0.76(0.44–1.30)	0.3191	1.49 (0.57–4.01)	0.4121

The change (Δ) denotes post antiviral treatment (post; M6) minus baseline (pre; baseline) values of Let-7 level based on the log _10_2^−Ct (Ct _target_—Ct_cel39_) method. The cut-off value was determined by the median value from the distribution of the let-7 expression on the log_10_2^−delta Ct method. Adjusted by age, sex, HCV genotype, and FIB-4 ≥ 3.25; * statistically significant at *p* ≤ 0.05.

## Data Availability

The data presented in this study are available in this article.

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
