# Peer review of "Circulating Let-7 Family Members as Non-Invasive Biomarkers for Predicting Hepatocellular Carcinoma Risk after Antiviral Treatment among Chronic Hepatitis C Patients"

_cancers, 2022, doi:10.3390/cancers14082023_

Round 1
Reviewer 1 Report
Dear authors,
- What is the significance of negative correlations with the fibrosis score in the present manuscript.
- Why was the non-SVR group less likely to develop HCC?
- Figure S1 needs to be colored and must be good at visualization.
- How do the serum levels of the Let-7 family predict HCC risk in patients with CHC using univariate and multivariate Cox’s proportional hazards model?
- What is the age of diagnosis for HCC at advanced stages?
Author Response
- What is the significance of negative correlations with the fibrosis score in the present manuscript.
We thank you for the comment. FIB-4 was defined by the following equation: (age × AST [U/L]) / (PLT [109/L]×ALT [U/L]1/2); this detail has been provided in the manuscript (lines 107 and 108). We also observed meaningful positive correlations between platelet count (PLT) and Let-7a (r=0.2989, p<0.0001), let-7b (r=0.3171, p<0.0001), let-7d (r=0.4028, p<0.0001), let-7e (r=0.3096, p<0.0001), let-7f (r=0.2833, p<0.0001), let-7g (r=0.2595, p<0.0001), let-7i (r=0.3847, p<0.0001), and miR-98 (r=0.3015, p<0.0001) (lines 185–188). Thus, a negative correlation between the FIB-4 score and Let-7 family members (except let-7c) was observed. It indicated that down-regulation of circulating let-7 family was associated with advanced fibrosis, which is a risk factor predictive of HCC development.
- Why was the non-SVR group less likely to develop HCC?
Non-SVR was associated with more likely to develop HCC. In the current study, 26.8% (26/97) of non-SVR patients developed HCC, compared to 21.5% (28/130) of SVR patients developed HCC. The difference did not reach significance is because that the two groups were not enrolled in a consecutive manner. Instead, we enrolled more SVR patients with HCC developed, so that it’s more feasible for the current study to evaluate the role of let-7 family in the HCC risk after antiviral therapy.
- Figure S1 needs to be colored and must be good at visualization.
We very appreciate for this suggestion. We have replaced with high resolution image.
- How do the serum levels of the Let-7 family predict HCC risk in patients with CHC using univariate and multivariate Cox’s proportional hazards model?
We thank you for the comment. We have shown the Let-7 family members predict HCC risk in patients with CHC using univariate and multivariate (Table 4). We have provided the results of Kaplan-Meier survival analysis of let-7 family cluster 1 (let-7a, let-7d, let-7e, and let-7g) and cluster 3 (let-7c, let-7f, and miR-98) with HCC development in the SVR and non-SVR groups in Supplement Figure S2 and Figure S3 (lines 339–353). We have mentioned this in the Discussion (lines 248–251).
- What is the age of diagnosis for HCC at advanced stages?
We thank you for the comment. The age of patients at the diagnosis of advanced HCC (BCLC-C stage) was 67.89 ± 9.46 years. We have provided the information of the 54 de novo HCC patients in the Materials and Methods as Table S1 (lines 102 and 321–324).

Reviewer 2 Report
It is an interesting manuscript about “Circulating let-7i Expression as Non-invasive Biomarker for Predicting Hepatocellular Carcinoma Risk After Treatment among Chronic Hepatitis C patients”.
My concern is determined in the following points.
Approximately 90% of hepatocellular carcinoma (HCCs) develop in people with risk factors such as cirrhosis or noncirrhotic chronic hepatitis. HCV is the most cause of HCC in Western countries. Successful DAA therapy is associated with a 71% reduction in HCC risk. However, patients with cirrhosis continue to have a significantly elevated risk of HCC despite achieving SVR, with HCC being reported even 10 years after SVR. It is of critical importance to continue HCC surveillance in those with cirrhosis who have achieved SVR. DAA therapies have led to a significant reduction in HCC incidence from 3.6% per year to 1.8% per year. Those without cirrhosis have a lower HCC incidence of 1% per year. Those without cirrhosis but with a Fib-4 score >3.25 had an incidence rate for HCC of 0.9% per 100 person-years.
The best strategy for surveillance is the combination of alpha-fetoprotein and ultrasound of the liver every 6 months. The Liver imaging Reporting and Data System (LI-RADS) achieves higher than 95% positive predictive value for the noninvasive diagnosis of HCC on computed tomography (CT), MRI and contrast-enhanced ultrasound (CEUS). Cirrhosis is found in more than 90% of individuals diagnosed with HCC. Thus, any cause of chronic liver disease and ultimately cirrhosis should be considered as the main risk factor for HCC. AFP-L3 and DCP are complementary to AFP, as they may be abnormal even when AFP is not abnormal. The combination of AFP, AFP-L3, and DCP had greater predictive power than did any individual marker.
In this study showed that circulating Let-7 family levels were measured at the initiation of antiviral treatment and 6 months after the end of treatment.
Circulating let-7i can be used for early CHC surveillance with HCC risk after antiviral treatment. When did patients develop HCC and what were characteristics of HCC such as size and numbers after antiviral treatment? This study had several limitations. AFP levels were not available after anti-viral treatment. Therefore, this study cannot compare the advantages and disadvantages of surrogate biomarkers as mentioned above.
Above mentioned should be referred to.
Author Response
When did patients develop HCC and what were characteristics of HCC such as size and numbers after antiviral treatment?
We thank you for the comment. We have provided the information of the 54 de novo HCC patients in the Materials and Methods as Table S1 (lines 102 and 321–324). The age of patients at the diagnosis of advanced HCC (BCLC-C stage) was 67.89 ± 9.46 years after antiviral treatment. The increasing tumor size was associated with BCLC stage.
We totally agree with the great comments that post-treatment AFP is associated with risk of HCC development. Also, the combination of AFP, AFP-L3, and DCP had greater predictive power than did any individual marker. Unfortunately, we did not collect the post-treatment AFP data in the current study. DCP was not routinely used in Taiwan before 2020. AFP-L3 is not available clinically in Taiwan. We will address the issues as limitation of the current study (lines 288-291).

Round 2
Reviewer 1 Report
I like the revised manuscript and no further comments.